# The Seismic Performance of New Self-Centering Beam-Column Joints of Conventional Island Main Buildings in Nuclear Power Plants

**DOI:** 10.3390/ma15051704

**Published:** 2022-02-24

**Authors:** Qiang Pei, Cong Wu, Zhi Cheng, Yu Ding, Hang Guo

**Affiliations:** 1College of Architectural Engineering, Dalian University, Dalian 116622, China; wucforever@163.com (C.W.); wateryu147@163.com (Y.D.); gh897422078@163.com (H.G.); 2Department of Construction Engineering Management, Southwest Jiaotong University Hope College, Chengdu 610400, China; czh9182021@163.com

**Keywords:** shape memory alloy, self-centering, beam-column joints, seismic performance

## Abstract

In order to improve the deformation energy consumption and self-centering ability of reinforced concrete (RC) frame beam-column joints for main buildings of conventional islands in nuclear power plants, a new type of self-centering joint equipped with super-elastic shape memory alloy (SMA) bars and a steel plate as kernel components in the core area of the joint is proposed in this study. Four 1/5-scale frame joints were designed and manufactured, including two contrast joints (a normal reinforced concrete joint and a concrete joint that replaces steel bars with SMA bars) and two new model joints with different SMA reinforcement ratios. Subsequently, the residual deformation, energy dissipation capacity, stiffness degradation and self-centering performance of the novel frame joints were studied through a low-frequency cyclic loading test. Finally, based on the OpenSees finite element software platform, an effective numerical model of the new joint was established and verified. On this basis, varying two main parameters, the SMA reinforcement ratio and the axial compression ratio, a simulation was systematically conducted to demonstrate the effectiveness of the proposed joint in seismic performance. The results show that replacing ordinary steel bars in the beam with SMA bars not only greatly reduces the bearing capacity and stiffness of the joint, but also makes the failure mode of the joint brittle. The construction of a new type of joint with consideration of the SMA reinforcement and the steel plate can improve the bearing capacity, delay the stiffness degradation and improve the ductility and self-centering capability of the joints. Within a certain range, increasing the ratio of the SMA bars can further improve the ultimate bearing capacity and energy dissipation capacity of the new joint. Increasing or decreasing the axial compression ratio of column ends has little effect on the overall seismic performance of new joints.

## 1. Introduction

At present, nuclear power plants have adopted higher seismic design criteria according to the seismic hazard evaluation of the site. For example, the seismic design criteria adopted by Tianwan, Taishan, and Haiyang nuclear power plants in China are 0.2 g, 0.25 g, and 0.3 g, respectively, which are higher than the local seismic level [1]. The safety of domestic and foreign nuclear power plants using this kind of seismic design standard has been verified and affirmed in previous conventional earthquakes. However, for the super-design reference earthquake, the current response measures are mainly to improve the seismic isolation design of nuclear power engineering structures and to analyze and evaluate the nuclear power system through seismic margin assessment and seismic probabilistic risk assessment [2,3]. The above methods cannot be separated from the judgment of empirical data, and the evaluation results have great uncertainty. Especially in the case of two or more unknown super-benchmark accidents, such as accidental damage to buildings adjacent to the nuclear island and secondary disasters caused by beyond-design basis earthquakes, as in the Fukushima nuclear accident in 2011, such a beyond-design basis earthquake may still pose a greater security threat to nuclear power engineering. However, the structural design considering the superposition of multiple disasters, such as earthquakes and tsunamis, is neither economical nor convenient. On the other hand, under the requirements of the current economic sustainable development, the research on seismic engineering has gradually developed from seismic isolation to the direction of recoverable function [4]. In this trend, higher requirements are also put forward for major projects represented by nuclear power projects that are related to the national economy, people’s livelihoods and the national economic lifeline, so as to achieve the seismic goal of the function not being interrupted or restored as soon as possible during the earthquake, and the normal use can be achieved without repair or with only slight repair after the earthquake. At the same time, a large number of seismic damage investigation results [5,6,7,8,9] show that beam-column joints, as an important hub for coordinating deformation and transfer load distribution in frame structure systems, are also one of the most seriously damaged parts, especially in related nuclear power engineering frame structure buildings under beyond-design basis earthquake. The traditional seismic design improvement method of concrete beam-column joints is mainly strengthening stirrups or using high strength concrete [10,11,12,13]; at the same time, it brings about a substantial increase in the construction cycle and cost and does not break through the performance limitations of traditional building materials. It is difficult to meet the requirements of structural recoverable functions. Concrete-filled steel tubular columns can improve the bearing capacity and seismic performance of the structure to a certain extent, and in recent years, some progress has been made in the experimental and theoretical research of these columns [14,15,16,17], but their practical application in nuclear power engineering is still rare. At the same time, the super-elastic shape memory alloy (SMA) has been rapidly developed and applied due to its special material functional properties [18,19,20,21,22]. It also provides a new idea for improving safety in nuclear power frame structure engineering under the action of beyond-design basis earthquakes.

The shape memory alloy (SMA) is a new intelligent material that takes into account sensing and driving functions. When the external force is unloaded, the inverse phase transformation drive can automatically restore the strain up to 8%–10% instantaneously [23]. At present, it has been widely studied and applied in energy dissipation braces [24,25], isolation bearing [26,27], and various dampers [28,29,30]. Based on these spontaneous and instantaneous recoverable super-elastic characteristics, the improved replacement between SMA bars and ordinary steel bars in conventional concrete beams and columns provides a new research and design idea for improving the mechanical properties of concrete beam-column joints. The research on the seismic performance of RC beam-column joints mainly explores the influence of other structural members, such as wide beams and slabs [31,32], and reinforcement methods, such as FRP and BFRP reinforcement [33,34,35,36]. However, there are few experimental studies on the seismic performance of new self-centering concrete beam-column joints based on SMA tendons [37,38,39], and the design structure of the new joints in the related research reported is relatively larger than that of the traditional concrete joints, which lacks the simplicity and practicability suitable for actual construction.

Based on this, in order to further promote the application of SMA reinforcement in the field of structural engineering and optimize the energy consumption and self-recovery ability of important concrete frame engineering structures, considering that the frame structure edge joints in earthquake damage are often more serious than the internal joints, our research group designed a new type of self-centering and low-damage joint with the conventional frame edge joint of conventional island main buildings in nuclear power plants. Different from the previous related self-centering nodes, in order to enhance the application feasibility, the structural design is closer to the traditional steel binding process. It is proposed to study the failure process, hysteretic characteristics, energy dissipation capacity, stiffness degradation, and self-centering capacity of the new joint through experiment and numerical simulation, so as to provide a certain basis for the practical application of the new joint in important frame structures such as conventional island main buildings in nuclear power plants.

## 2. Experimental Program

### 2.1. Test Specimens

Four frame beam-to-column joints with a scale ratio of 1:5 were designed and manufactured, including two self-centering new beam-to-column joint models numbered by PSJD-1 and PSJD-2 (ordinary longitudinal reinforcement plus hybrid joints with different diameters of SMA reinforcement) and two comparative joints (ordinary reinforced concrete beam-to-column joints numbered by PJD-1, SMA reinforced beam-to-column joints numbered by SJD-1). The geometric dimensions of each specimen were the same and were made according to the current concrete design specifications in China. The fixed steel plate required in this experiment not only played the role of connecting shape memory alloy rods, but also played the role of a longitudinal reinforcement elbow. In order to reduce the adverse effect on the anchorage connection end when the joint was damaged and cracked, two steel plates were placed 50 mm away from the outer edge of the beam and the column, respectively. The size reinforcement and specific parameters of the specimen are shown in Figure 1 and Table 1. The steel end plate of the new joint has reserved holes for SMA bars and ordinary steel bars to pass through. The SMA bars and the steel end plate were connected by bolt anchorage, and the ordinary steel bar and the steel end plate were welded. The connection structure is shown in Figure 2.

### 2.2. Materials

#### 2.2.1. SMA Materials

The Ni–Ti shape memory alloy bar used in this test was customized in Baoji Long Qiangfeng Titanium Industry Co., Ltd (Baoji, China). The chemical composition was approximately as follows: Ni: 54.38%; Ti: 45.575%; others: 0.045%. The length and diameter of the fabricated dumbbell-shaped SMA bar specimens were 130 mm and 10 mm, respectively.

Studies have shown that [37,40] heat treatment can greatly improve the super-elasticity of SMA bars. The heat treatment process used in this experiment was as follows: the SMA specimens were placed in a high-temperature furnace with a constant temperature of 400 °C for 25 min, and the water was taken out immediately after the end. In order to further stabilize the mechanical properties of SMA bars, the specimens after heat treatment were placed in boiling water and ice water for 3 min, and the above cold and hot cycles were repeated five times. The cyclic tensile test of SMA bars was carried out according to the strain amplitude (1%, 2%, …, 8%). The cyclic loading and unloading at all levels were carried out once, the strain rate was 0.0015 s^−^^1^, and the test room temperature was 25 °C. The constant temperature heating furnace for heat treatment and the tensile test loading device are shown in Figure 3. The properties of the Ni–Ti alloy after heat treatment are shown in Table 2.

#### 2.2.2. Steel Plate, Reinforced Steel, and Concrete Materials

The steel plate material used in this test were Q235 (ordinary carbon structural steel and had a nominal design yield strength of 235 MPa), and the specific size of the steel plate is shown in Figure 4. The beam column longitudinal steel bar selection model for HRB400 (its standard value of yield strength is 400 MPa), the stirrup selection of HPB300 (similar as HRB400, its standard value of yield strength is 300 MPa), and the mechanical properties of the steel parameters are shown in Table 3. The measured results of the compressive strength of the concrete cube (dimensions: 100 × 100 × 100 mm) test block are shown in Table 4.

### 2.3. Load Equipment

In order to be close to the actual stress state of the joint, the steel box was set at the lower end of the column, the spherical steel hinge was welded on the steel box in the specimen fabrication process, and the spherical hinge with a loading jack was used at the upper end of the column. At the same time, considering the influence of the beam-column self-weight, the specimen was installed by the vertical beam of the column in the horizontal direction, and the lateral support of the specimen was realized by connecting the scaffolding with the reaction wall. The schematic diagram of the test loading device is shown in Figure 5. Figure 6 shows the scene photos of the loading test. The effect of temperature on the performance of the joint is difficult to achieve, so all the tests were carried out at room temperature.

### 2.4. Layout of Test Points

The resistance strain gauges were affixed to the concrete surface of the longitudinal reinforcement, the SMA reinforcement, and the plastic hinge area of the beam-column member to measure the strain at the corresponding position. The strain gauges used on the steel bar inside the joint specimens were BX120–3AA, and those of the concrete surface were BX120–50AA. The strain gauge arrangement of the specimen is shown in Figure 7. Considering that the overall size of the component in this test was relatively small, and there was an inevitable slight vibration in the process of low cyclic loading, in order to measure the plastic hinge length of each specimen more accurately, the high-precision non-contact full-field strain measurement system (video image correlate-3d, VIC-3D, Correlated Solutions, Inc., Columbia, SC, USA) based on digital image technology was selected to replace the conventional displacement meter for measurement. The VIC-3D system can measure the full-field three-dimensional displacement and strain of the surface of the object under test and has low environmental requirements. In principle, clear images of the area under the test can be taken indoors and outdoors. The accuracy and feasibility of digital image correlation technology and VIC-3D based on this technology have been proven [41,42,43,44,45].

The measurement range was drawn in the beam-column connection area. Speckles with a diameter not less than 2 mm were arranged in the measurement range, and the correction plate with a side length of 20 mm was selected to correct. The camera was set up in place and focus, and the vertical displacement value Y at any position in the measurement range can be measured by the measurement system. According to the calculation needs, seven measuring points were selected along the beam end in this experiment, and the distance between each measuring point was 60 mm. The measurement range size and the specific location of the selected measuring points are shown in Figure 8. The vertical displacement of each point is recorded as Y_1_, Y_2_, …, Y_6_, Y_7_. If a plastic hinge is formed between the measuring points marked *n* (1 ≤ *n* ≤ 7) from the beam-column junction, the spacing of each measuring point after the measuring point along the beam length should be the initial 60 mm, and the vertical displacement of the next measuring point is denoted as Y_n+1_, and so on. If Δ1 = Y*_n_*_+1_ − Y*_n_*, Δ2 = Y*_n_*_+2_ − Y*_n_*, …, Δ*K* = Y*_n_*_+*K*_ − Y*_n_* (2 ≤ *K* ≤ 6), then Δ1, Δ2, Δ*K* should satisfy Equation (1) as follows:(1)$$\sin\alpha =\frac{\Delta 1}{60}=\frac{\Delta 2}{2\times 60}=\frac{\Delta 3}{3\times 60}=\cdots =\frac{\Delta K}{n\times 60}$$

Here, *α* is an angle which is between the axis of the undeformed beam outside the end point of the plastic hinge and the horizontal direction, as shown in Figure 9. The *n* value of Equation (1) is the plastic hinge end point. Considering the inevitable measurement error, the absolute value of the sinusoidal difference obtained by Equation (1) is not greater than 1.0 × 10^−4^. It can be approximately equal, and the solution schematic diagram is shown in Figure 9. The measured plastic hinge length calculated for each node is shown in Table 5.

### 2.5. Loading History

The vertical load was applied to the top of the column before cyclic loading, and the axial force was kept constant after the axial force was relatively stable. The experimental control of the axial compression ratio was 0.25, while the actual vertical load value was 445 kN. The low cyclic loading test of the joint specimen adopted the whole process displacement control method, and the loading rate was 0.2 mm/s. In this experiment, before the specimen yield, the increment of displacement grading loading was 1 mm, and each stage was recycled twice. After the specimen yield, the yield displacement was denoted as Δ, and it was then loaded by an integral multiple of Δ. Each stage was cycled twice. The loading was terminated when the load dropped to 85% of the ultimate load; otherwise, the component was damaged, as shown in Figure 10.

## 3. Results and Analysis

### 3.1. Test Phenomenon

#### 3.1.1. Comparison Test Piece PJD-1

Specimen PJD-1 was a comparative joint with ordinary steel bars at the beam end. When the loading displacement was 2.0 mm, the first oblique crack appeared in the middle of the beam, with a length of 6.45 cm and a width of 0.10 mm. At the same time, the vertical crack with a length of 4.3 cm and a width of 0.2 mm developed at the root of the beam. The load–displacement curve gradually deviated from the straight line, and the specimen entered the yield stage. When the loading control displacement was one times the yield displacement, the first small crack appears at the bottom of the beam (19 cm away from the core area of the node), and the crack length was 4.16 cm. When the loading control displacement was three times the yield displacement, the vertical cracks at the root of the beam and the oblique cracks in the middle of the beam were successively penetrated from up to down, and were developed in the surroundings, resulting in many small development cracks. At six times the yield displacement, the peak load of the specimen reached 54.72 kN; at 11 times the yield displacement, the load value of the specimen decreased from 50.08 kN to 37.45 kN, and the test ended. At this point, the vertical main crack at the root of the beam was relatively wide, the concrete at the upper and lower sides of the root was obviously crushed, and the concrete of the protective layer was stripped, as shown in Figure 11a.

#### 3.1.2. Comparison Test Piece SJD-1

The SJD specimen was the comparison node, where the longitudinal bars at the beam end were all SMA bars. When the loading displacement was 1.0 mm, the first vertical crack appeared at the root of the specimen, and the length of the crack was about 3.12 cm. When the reverse loading was carried out, the root crack had been quickly penetrated, and the crack width was about 0.3 mm. The load–displacement curve had deviated from the straight line, and the specimen entered the yield stage. When the loading control displacement was four times the yield displacement, the crack width increased, and a small amount of the surface concrete at the root of the beam fell off. At 17 times the yield displacement, a low ‘boom’ sound was heard, and the load value of the specimen reached 31.66 kN. When the load continued, the load value of the specimen immediately dropped to 16.27 kN, and the test ended. At this point, the vertical main crack at the root of the beam was wider than that at the PJD-1 node, but there was no obvious damage to the upper and lower concrete at the root of the beam, as shown in Figure 11b.

#### 3.1.3. Test Piece PSJD-1 and Test Piece PSJD-2

Both PSJD-1 and PSJD-2 are model test joints with SMA bars and steel endplates. Taking the specimen PSJD-1 as an example, the loading process of this new type of joint was introduced. When the loading displacement was 2.0 mm, the first vertical crack appeared at the root of the beam, with a length of about 9.5 cm and a width of about 0.1 mm. There was no obvious oblique crack in the middle of the beam. When the displacement was controlled to 3 mm, the load–displacement curve begun to deviate from the straight line, the crack width was 0.2 mm, and the crack length reached 10.60 cm; the specimen thus entered the yield stage. When the loading control displacement was three times the yield displacement, the crack width reached 2.0 mm, the residual crack width after unloading was 1.2 mm, and the cracks in the root and middle of the beam were successively penetrated. At seven times the yield displacement, the concrete at the root of the beam showed spalling. At ten times the yield displacement, the load value began to decrease with the low ‘boom’ sound inside the specimen (which is believed to be the connection fracture between the steel plate and the longitudinal reinforcement), and the final load value of the specimen decreased from 56.11 kN to 46.02 kN. At this point, the crack width of the vertical main crack at the root of the beam was significantly smaller than that of the PJD-1 node, and the compressive area and the stripping volume of the concrete at the upper and lower sides of the root were smaller than those of the PJD-1 node, as shown in Figure 11c.

The first half of the loading process of the PSJD-2 specimen was similar to that of the PSJD-1. When the loading displacement was 2.0 mm, the first vertical crack appeared at the root of the beam, with a length of about 12.0 cm and a width of about 0.1 mm. When the loading displacement was controlled to 4 mm, the root crack width reached 0.5 mm, and the root crack had been penetrated. The first crack appeared in the middle of the beam component, and the crack length was 22.45 cm. The specimen thus entered the yield stage. When the loading control displacement was three times the yield displacement, the core area of the joint specimen began to peel off the concrete block, the root crack width reached 2.0 mm, and the middle crack width reached 1.0 mm. However, when the load was four times the yield displacement, the abnormal sound, similar to that heard in the PSJD-1 experiment, was heard early and originated from inside the specimen, and the load value then immediately decreased from 81.78 kN to 65.80 kN. The specimen thus failed immediately. Due to the relatively early failure of the specimen, the node final crack was smaller than that of the PSJD-1 joint, and there were no obvious signs of concrete compression and spalling, as shown in Figure 11d.

### 3.2. Beam End Load–Displacement Hysteresis Curve

The hysteresis curves of each specimen in this test are shown in Figure 12.

(1)Comparing Figure 12a,c, it can be seen that the pinching phenomenon of the hysteretic curve of the PJD node is obvious. The hysteretic curve of the PSJD-1 node is rounder than that of the PJD-1 node, and the ultimate bearing capacity is increased by about 44%, indicating that the structural form of the built-in SMA reinforcement-steel end plate improves the hysteretic energy dissipation capacity of the node and significantly improves the bearing capacity of the node.(2)Although the ultimate bearing capacity of the SJD-1 node is lower than that of the PJD-1 node, the residual displacements at all levels before failure are much smaller. This shows that the SMA bar material can significantly improve the self-centering ability of the structure.(3)Although the PSJD-2 joint failed prematurely in this test, it can be seen, by comparing Figure 12a,c, that the residual deformation of the PJD-1 joint after complete unloading at each loading stage is very large. The hysteresis curve of the PSJD-1 node is obviously ‘flag-shaped’. After each loading stage is completely unloaded, the residual deformation of the node is very small, indicating that the new node has a good self-centering ability.

### 3.3. Skeleton Curve

As an important basis for restoring the force model and nonlinear seismic response analysis, the skeleton curve reflects the deformation, energy consumption, and stiffness degradation of specimens in different stages. The skeleton curve of each node in this test is shown in Figure 13. As the failure of the PSJD-2 specimen is relatively early, this paper mainly analyzes the skeleton curves of the other three specimens.

The following conclusions can be drawn from Figure 13.

(1)The ultimate bearing capacity of the PSJD-1 node and the area surrounded by the skeleton curve and the abscissa axis are larger than those of the PJD-1 node. Although the PSJD-2 node fails earlier, the ultimate bearing capacity before failure and the area surrounded by the abscissa axis are further improved compared to those of the PSJD-1 node. This shows that the ultimate bearing capacity and the energy dissipation capacity of the joint can be significantly improved by adding a super-elastic SMA reinforcement-steel end plate and by increasing the reinforcement ratio of the corresponding SMA reinforcement.(2)In the initial elastic stage, the skeleton curve slope of the PJD-1 node is much larger than that of the SJD-1 node, indicating that SMA material will reduce the initial stiffness of the component under the same longitudinal reinforcement ratio. This is because the Young’s modulus of the SMA bars (65.4 GPa) is much smaller than that of the steel bars (203 GPa).(3)In the yield stage, the yield displacement value of the PSJD-1 joint is significantly larger than that of the PJD-1 joint, and the bearing capacity of the PJD-1 joint and the SJD-1 joint decreases rapidly after yield. However, the bearing capacity of the new joint PSJD-1 is relatively stable after yield and can withstand relatively larger deformation, indicating that the built-in super-elastic SMA reinforcement-steel end plate can delay the yield of the joint to a certain extent and improve the ductility and damage resistance of the joint.

### 3.4. Residual Displacement of the Beam End

The residual displacement of the beam end refers to the residual plastic deformation of the beam end after unloading, and its value can reflect the self-centering ability of the component. The beam end loading-residual displacement curve of each joint specimen is shown in Figure 14.

It can be seen in the graph that the residual displacement of the three nodes with the built-in super-elastic SMA tendons is less than that of the PJD-1 node under the same loading level. When the peak displacement of the beam end is 30 mm, the recoverable deformation of the PJD-1 joint is 25%, while that of the PSJD-1 joint is 57% (the residual deformation of the beam end is 13 mm); due to the poor anchorage between the SMA bars and the concrete, the failure of the SJD-1 joint occurs earlier (the maximum loading displacement of the beam end is only 20 mm), but the residual displacement of the SJD-1 joint is still less than that of the PJD-1 joint. This shows that, under a certain loading amplitude, the built-in SMA reinforcement can significantly improve the self-centering ability of the node.

### 3.5. Energy Consumption Capacity

The structure dissipates the ground motion energy mainly through plastic deformation, and the seismic performance of the structure can be reflected by the energy dissipation capacity. The energy dissipation value of each specimen under different load displacement can be obtained by calculating the enclosed area of the hysteresis curve in Figure 12, as shown in Figure 15. Figure 15 shows the energy dissipation curve of the four node specimens.

The figure shows the following:(1)When the specimen is in the elastic stage, the steel bars and SMA bars in the joint do not yield, and the energy dissipation capacity of each joint is very low.(2)When the specimen is in the plastic stage, the slope of the displacement–energy curve at the beam end of the PSJD-1 node and the PSJD-2 node is greater than that of the PJD-1 node, and the slope of the SJD-1 node is the smallest, indicating that the energy dissipation capacity of the new node is better than that of the ordinary node, while the energy dissipation capacity of the SJD-1 node is the weakest. When the beam end displacement exceeds 20 mm, the energy dissipation capacity of the PJD-1 node tends to be stable, while the energy dissipation of the PSJD-1 node continues to increase. This shows that the built-in SMA rib-steel end plate can effectively improve the energy dissipation capacity of the joint under large displacement deformation.(3)Comparing the PSJD-1 and PSJD-2 joints, it can be seen that, with an increase in the SMA reinforcement ratio, the energy dissipation capacity of the new joints is further improved.

### 3.6. Stiffness Degradation

We selected the ring stiffness K as an index to evaluate the stiffness degradation of the beam-column joints. The obtained stiffness–ductility coefficient curve is shown in Figure 16.

The following conclusions can be drawn from Figure 16:(1)Comparing the initial stiffness of the four nodes, it can be seen that the initial stiffnesses of the PSJD-1 node, the PSJD-2 node, and the PJD-1 node show little difference. The initial stiffness of the SJD joints is obviously smaller than that of the PJD-1 joints, because the Young’s modulus of the SMA bars is much smaller than that of the steel bars.(2)When the ductility factor is less than 1, the stiffness degradation rate of the SJD-1 joint is the fastest; this is because the surface of the SMA reinforcement is relatively smooth, and the bonding force between the SMA reinforcement and the concrete is low. After the concrete cracks, the penetrating cracks are quickly formed, and the stiffness decreases rapidly. When the ductility coefficient is greater than 1, the stiffness degradation of the SJD-1 joint is limited; this is because that with a large amount of concrete cracking out of work, the stiffness ratio increases, and the SMA tendons have excellent super-elasticity.(3)Comparing the PSJD-1 node, the PSJD-2 node, and the PJD-1 node shows that, when the ductility coefficient is less than 1, the stiffness degradation rates of the PSJD-1 and PSJD-2 nodes are significantly smaller than those of ordinary nodes. When the ductility coefficient is greater than 1, although the premature failure of the internal components of the PSJD-2 node causes the stiffness of the node to rapidly degrade, and thus loses comparability, the stiffness of the PSJD-1 node is greater than that of the ordinary node, and the degradation rate is slower. This shows that the stiffness degradation rate of the node can be delayed with a built-in SMA reinforcement-steel end plate, so that the lateral displacement margin of the structure is larger, which can effectively improve the seismic performance of the structure.

## 4. Numerical Simulation

### 4.1. Model Establishment and Verification

#### 4.1.1. Constitutive Model of SMA Material

The self-centering ’double flag’ constitutive model developed by Ferdinando Auricchio [46] based on OpenSees is selected, as shown in Figure 17. The constitutive description of the SMA material is relatively complex, and this model simplifies the constitutive curve of the material into three stages: elasticity, phase transformation and hardening. The corresponding rules are as follows:In the elastic stage, when the stress does not exceed $\sigma_{y}$, the loading and unloading processes develop linearly with stiffness $k_{1}$.In the phase transformation stage, when the stress value exceeds $\sigma_{y}$, continuous loading develops with $k_{2}$ as the stiffness, and this stage ends when the strain reaches $\varepsilon_{b}$; when unloading in this stage, the stiffness decreases linearly with *k*_1_ and then decreases linearly with $k_{2}$.In the hardening stage, when the strain value is such that $\varepsilon_{b}$ ≤ *ε* ≤ $\varepsilon_{\mu }$, the loading and unloading processes develop along a straight line with slope *r* × *k*_1_. When the unloading strain value is less than $\varepsilon_{b}$, the continuous unloading is the same as the (1)(2) stage. In general, the strain value does not exceed $\varepsilon_{b}$.

The definition of this constitutive model in OpenSees is implemented with seven parameters. Parameter values are shown in Table 6.

#### 4.1.2. Constitutive Model of the Steel Bar and the Steel Plate

The constitutive model of the steel bar and steel plate shown in Figure 18 is the Steel 02 model, proposed by Menegotto and Pinto [47], and modified by Filippou [48]. The model not only considers isotropic strain hardening, but also reflects the influence of the Bauschinger effect. It has a high computational efficiency and good numerical stability.

The constitutive relationship expressions of the Steel 02 model are shown in Equations (2)–(5):(2)$$\sigma_{eq}=b\varepsilon_{eq}+\frac{\left(1-b\right)\varepsilon_{eq}}{{\left(1+\varepsilon_{eq}^{R}\right)}^{\frac{1}{R}}}$$
(3)$$\sigma_{eq}=\frac{\sigma -\sigma_{r}}{\sigma_{0}-\sigma_{r}}$$
(4)$$\varepsilon_{eq}=\frac{\varepsilon -\varepsilon_{r}}{\varepsilon_{0}-\varepsilon_{r}}$$
(5)$$R=R_{0}-\frac{a_{1}\xi }{a_{2}+\xi }$$
where $\sigma_{\mathrm{eq}}$ and $\varepsilon_{\mathrm{eq}}$ represent the normalized stress and strain; $\sigma_{0}$ and $\varepsilon_{0}$ represent the stress and strain of the rebar in the initial state; $\sigma_{r}$ and $\varepsilon_{r}$ represent the stress and strain of the rebar at the yield point; $a_{1}$ and $a_{2}$ represent the curvature degradation coefficients; R and $R_{0}$ represent the transition curve curvature coefficient and the initial curve curvature coefficient; b and ξ represent the hardening coefficient of reinforcement and the plastic strain in the semi-periodic cycle.

#### 4.1.3. Theoretical Model of Concrete

There are three kinds of concrete constitutive models commonly used in OpenSees: the Concrete 01 model with zero tensile strength, the Concrete 02 model with linear tensile softening, and the Concrete 03 model with nonlinear tensile softening [48]. For the concrete at the joint, its tensile strength can be ignored from a macro perspective. Therefore, for this simulation we selected Concrete 01 as the constitutive model of concrete, as shown in Figure 19.

In the Concrete 01 model [49], the constitutive relation expression is shown in Equations (6)–(10):(6)$$\mathrm{when}\;\varepsilon \;<\;\varepsilon_{0},\;f=Kf_{c}\left[\frac{2\varepsilon_{c}}{\varepsilon_{0}}-{\left(\frac{\varepsilon_{c}}{\varepsilon_{0}}\right)}^{2}\right]$$
(7)$$\mathrm{when}\;\varepsilon_{0}\leq \varepsilon \leq \varepsilon_{c},\;f=Kf_{c}\left[1-Z\left(\varepsilon -\varepsilon_{0}\right)\right]$$
(8)$$\mathrm{when}\varepsilon \;>\;\varepsilon_{c},\;f=0.2Kf_{c}$$
(9)$$\varepsilon_{0}=0.002K$$
(10)$$Z=\frac{0.5}{\frac{3+0.29f_{c}}{145f_{c}}-0.002K}$$
where $\sigma_{c}$ and $\varepsilon_{c}$ represent the stress and strain of the concrete; $f_{c}$ is the compressive strength of the concrete cylinder; K is the strain increase coefficient caused by the constraint; Z is the slope of the strain drop segment; $\varepsilon_{0}$ is the peak pressure and strain of the concrete.

#### 4.1.4. Types of Beam-Column Elements

According to the actual situation of the test, the nonlinear beam-column element (nonlinear beam column) was selected for simulation. The element is characterized by allowing the stiffness to change along the length of the rod. At the same time, the resistance and stiffness matrix of the control section can be determined. In the nonlinear simulation, it has the advantages of fast convergence and high effectiveness [49].

#### 4.1.5. Fiber Model

The fiber model, with a wide application and a high accuracy, was selected for the beam-column section. The fiber model based on the assumption of the plane section divides the section of the component into a certain number of grids. The center of each grid is regarded as a numerical integration point. The longitudinal micro-section of the grid is defined as fiber. The section is divided into several types of fiber bundles, such as confined and non-confined concrete, and steel bars. By calculating the strain stress of each fiber, the stiffness of the whole section is obtained [50].

### 4.2. Model Validation and Parameter Analysis

#### 4.2.1. Model Validation

According to the test results of material properties, various parameters in the finite element model were set. Based on the actual displacement of each cycle in the test, the displacement-controlled loading simulation analysis of the model was carried out. The hysteresis curves and skeleton curves of the ordinary concrete joint PJD-1 and the new type of joint PSJD-1 with internal SMA reinforcement were compared with the test results, as shown in Figure 20 and Figure 21.

Figure 20 and Figure 21 show that the hysteresis curve and the skeleton curve obtained by simulation are basically consistent with the experimental results, and the established model well reflects the self-centering energy dissipation characteristics of the new type of joints. The peak value of forward loading is slightly smaller than that of the test results. Due to the relatively ideal state of the model in the numerical simulation, and many factors such as constraints and measurement on the test site, there are certain differences. However, from the overall perspective, the obtained hysteresis and skeleton curves still well verify the accuracy and effectiveness of the finite element model.

#### 4.2.2. Parameter Analysis

In order to further study the new frame joints to further optimize the design, considering that the PSJD-2 joints failed earlier in the test and failed to fully reflect the influence of SMA reinforcement ratio on the related performance of the new joints, based on the verified finite element analysis model, the parameters such as the reinforcement ratio and the axial compression ratio of the built-in SMA bars in the new joints were simulated and analyzed to further quantify their effects on the mechanical properties, such as hysteretic performance, energy dissipation, and the self-centering ability of the new joints.

Based on the conventional ordinary reinforced concrete frame joints, under the same built-in SMA reinforcement mode, the yield strength of the SMA bars is controlled to be the same, and three new joints (XJD) with different SMA bar diameters were considered, namely, XJD-1 with a diameter of 8 mm, XJD-2 with a diameter of 10 mm, and XJD-3 with a diameter of 12 mm. The diameter of the SMA bars was controlled to be the same (10 mm), and three kinds of SMA bars with different yield strengths were considered to study their effects on the seismic performance of the new joint (XJD): XJD-4 had a yield strength of 300 MPa, XJD-2 had a yield strength of 400 MPa, and XJD-5 had a yield strength of 500 MPa. Specific parameter analysis design is shown in Table 7.

Effect of the SMA Reinforcement Ratio

Figure 22, Figure 23, Figure 24 and Figure 25 show that, under the condition of suitable reinforcement, the bearing capacity of the new joint also increases significantly with the increase in the diameter of SMA reinforcement. The S-shaped skeleton curve shows that the new joint has good energy dissipation and ductility. With the increase in the diameter of the SMA bars, the overall stiffness increases, but the increase is small, because the Young’s modulus of the SMA bars is smaller than that of ordinary steel bars, and the influence of ordinary steel bars on the overall stiffness is small.

In the initial state, increasing the diameter of SMA reinforcement has little effect on the residual deformation of the new node. With the increase in the number of cycles, the final residual deformation of XJD-3 is reduced by nearly 28% compared with that of XJD-1, indicating that, under the premise of suitable reinforcement, increasing the diameter of SMA reinforcement can significantly improve the self-centering performance of the new node.

2.Effect of Different Axial Pressure Ratio

Figure 26 clearly shows that, when the other parameters of the joint model are exactly the same, within the allowable range, only the axial force applied to the column is changed, and the influence on the seismic performance of the whole joint is almost zero. The main object of the study is the core area of the joint, and the variable parameter object of the study is the beam member; therefore, when the axial compression ratio of the column is changed, there is no obvious effect on the mechanical performance of the whole joint.

## 5. Conclusions and Discussions

In this research, a new type of seismic-resisting self-centering beam-column joint with built-in SMA reinforcement and a steel end plate was designed and investigated experimentally and numerically based on a conventional beam-column edge joint of conventional island main buildings in a nuclear power plant. In particular, the seismic performance of the joint was validated through a series of low cyclic loading tests on four models. Good agreement was observed in the comparisons between the experimental and numerical results. The main conclusions and discussions are as follows:(1)The self-centering performance of the joint can be improved by adding super-elastic SMA reinforcement in conventional concrete joints. However, replacing all the longitudinal reinforcement at the beam end with SMA reinforcement not only greatly reduces the bearing capacity and stiffness of the joint, but also makes the failure mode of the joint brittle.(2)The structural form of the built-in SMA reinforcement-steel end plate can significantly improve the bearing capacity of the joint and improve the cracking damage degree of the joint, so as to improve the post-earthquake reparability of the joint.(3)The stiffness degradation of the joint can be delayed by using the built-in SMA reinforcement-steel end plate structure, and the joint has good displacement ductility and self-centering energy dissipation performance. When the peak displacement of the beam end is 30 mm, the PSJD-1 joint can recover the deformation up to 57%.(4)For the new joint constructed with built-in SMA reinforcement and a steel end plate, under the premise of suitable reinforcement, increasing the reinforcement ratio of the SMA reinforcement within a certain range can further improve the bearing capacity and self-centering energy dissipation performance of the joint.(5)Within a certain range, increasing or reducing the axial compression ratio at the column end has little effect on the overall seismic performance of the new joint.(6)It is worth noting that the above conclusions are obtained by experiments at room temperature. The tested joints embedded with an Ni–Ti SMA bar are suitable for use in relatively stable environments at room temperature. However, Ni–Ti alloys may not be suitable for outdoor applications because of their extreme sensitivity to temperature. In order to address this limitation, on the one hand, experimental or numerical simulation research on the influence of temperature sensitivity on test results should be carried out. On the other hand, other types of hyperelastic materials, such as monocrystalline materials with a large pseudoelasticity change limit, a high energy dissipation and excellent low-temperature properties, should also be considered for outdoor applications.

## Figures and Tables

**Figure 1 materials-15-01704-f001:**
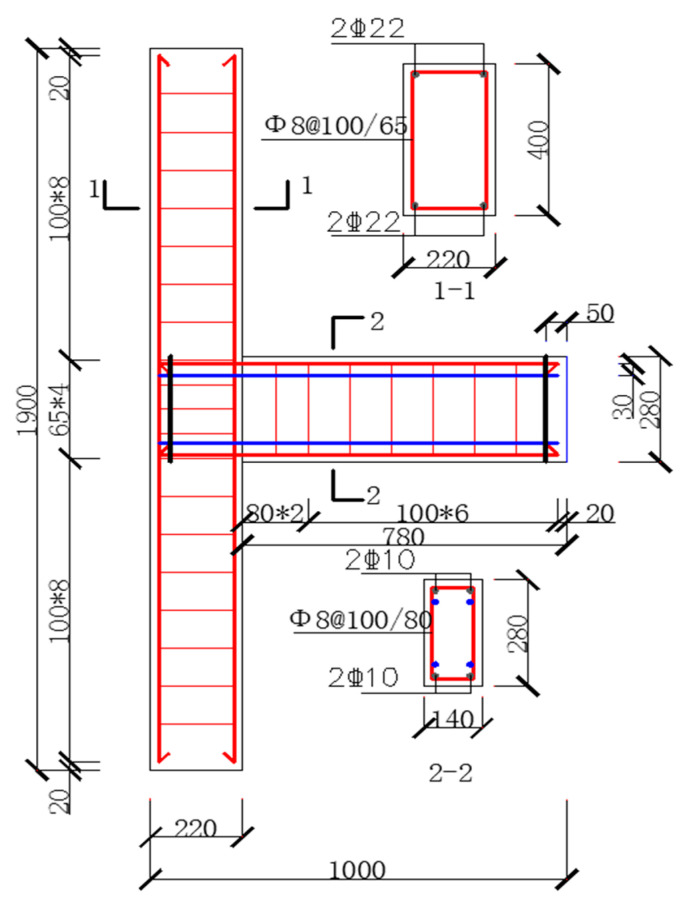
Specimen size and reinforcement details (unit: mm).

**Figure 2 materials-15-01704-f002:**
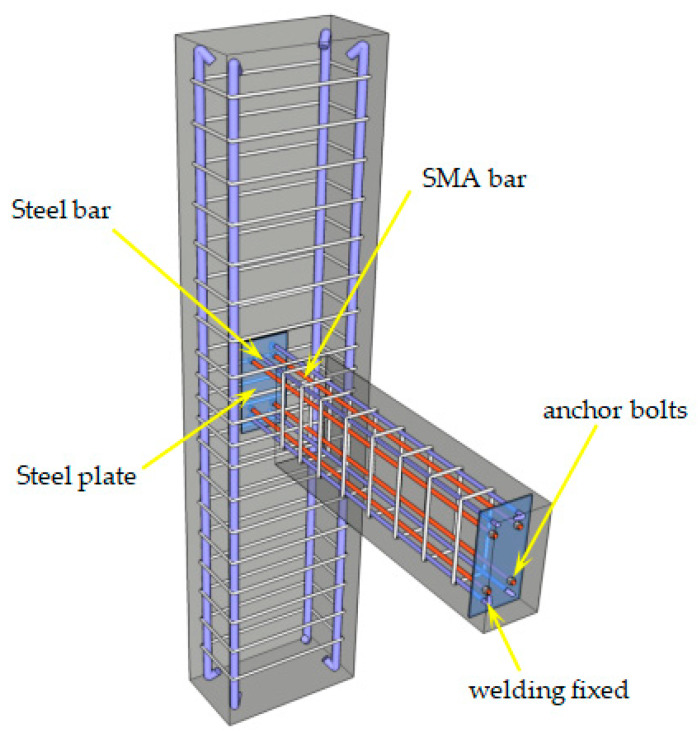
Three dimensional (3D) sketch of the self-centering joint.

**Figure 3 materials-15-01704-f003:**
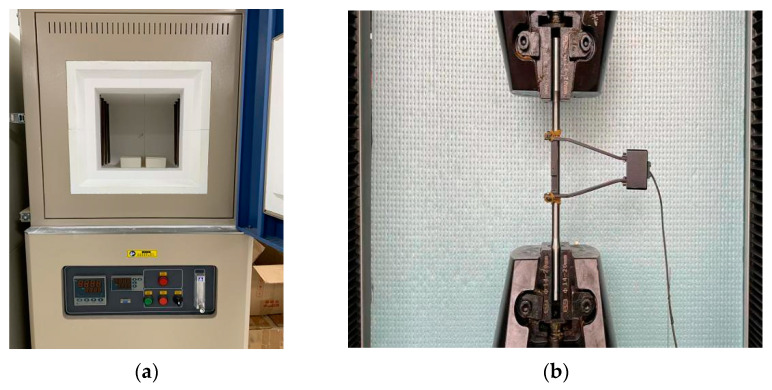
Heating and loading test device: (**a**) Thermostatic heating furnace; (**b**) SMA rod tensile test device.

**Figure 4 materials-15-01704-f004:**
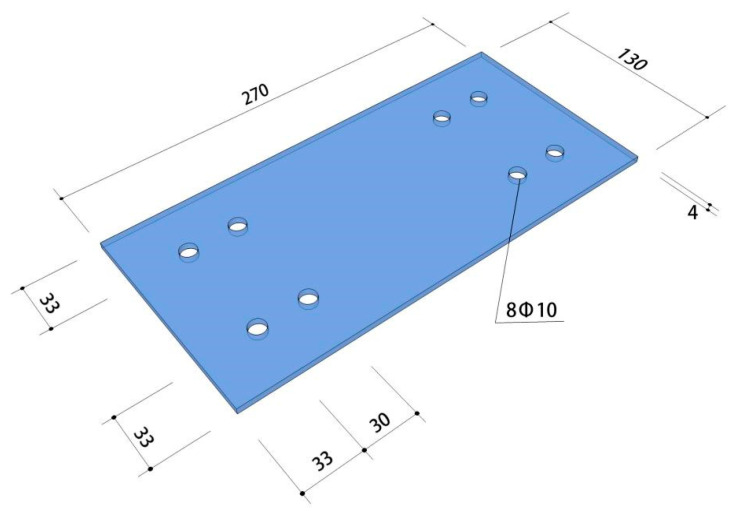
Dimensions of the steel plate (unit: mm).

**Figure 5 materials-15-01704-f005:**
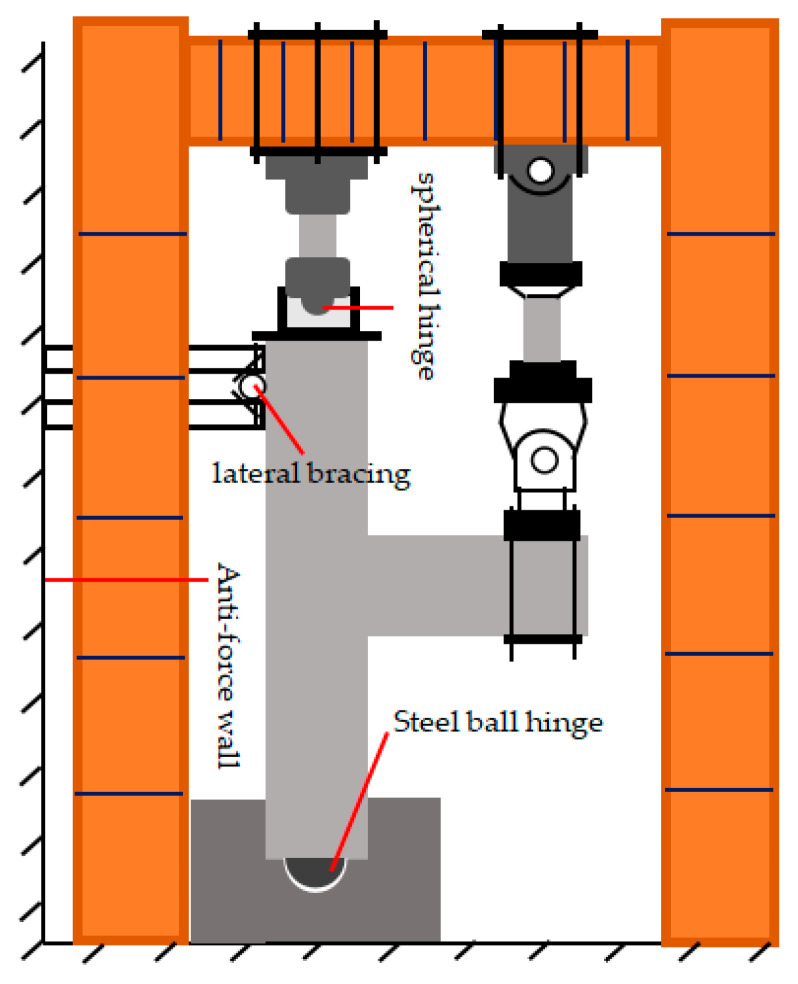
Schematic diagram of the test setup.

**Figure 6 materials-15-01704-f006:**
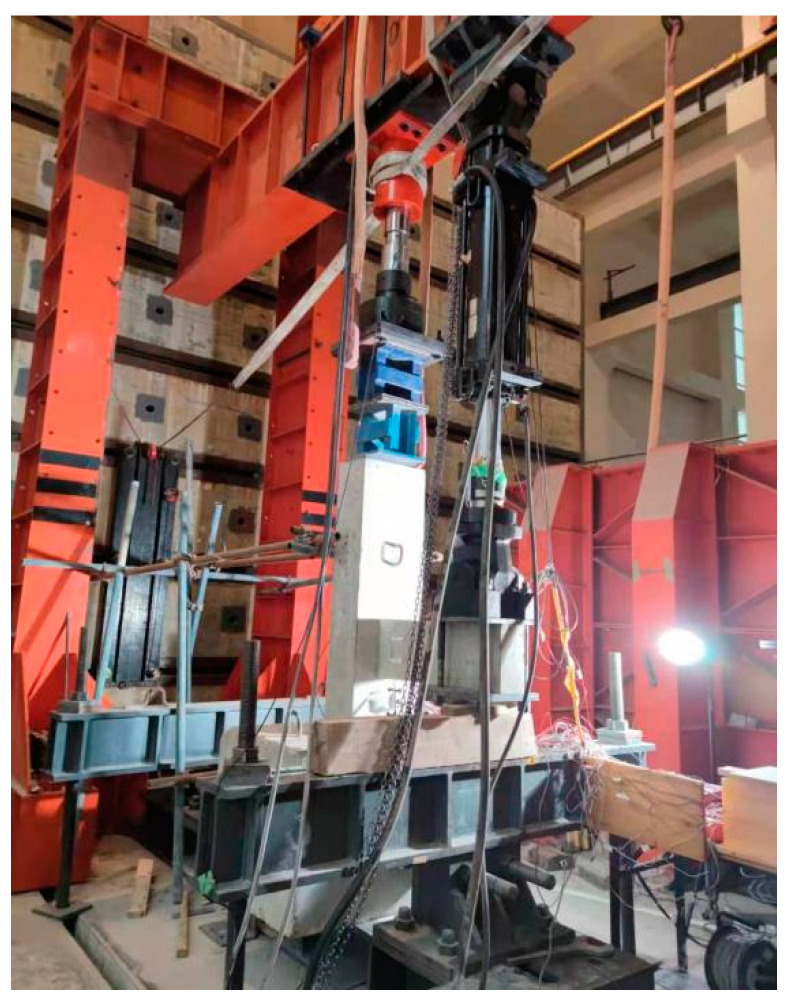
Photo of the test setup.

**Figure 7 materials-15-01704-f007:**
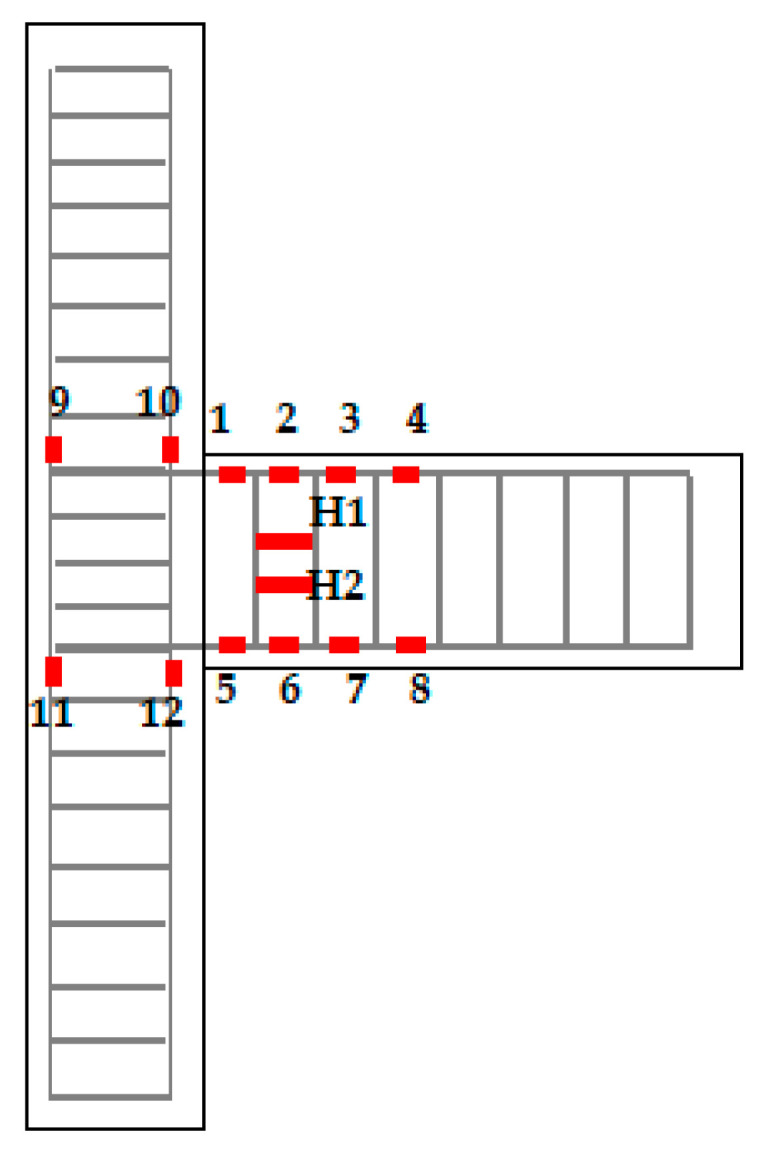
Strain gauge arrangement.

**Figure 8 materials-15-01704-f008:**
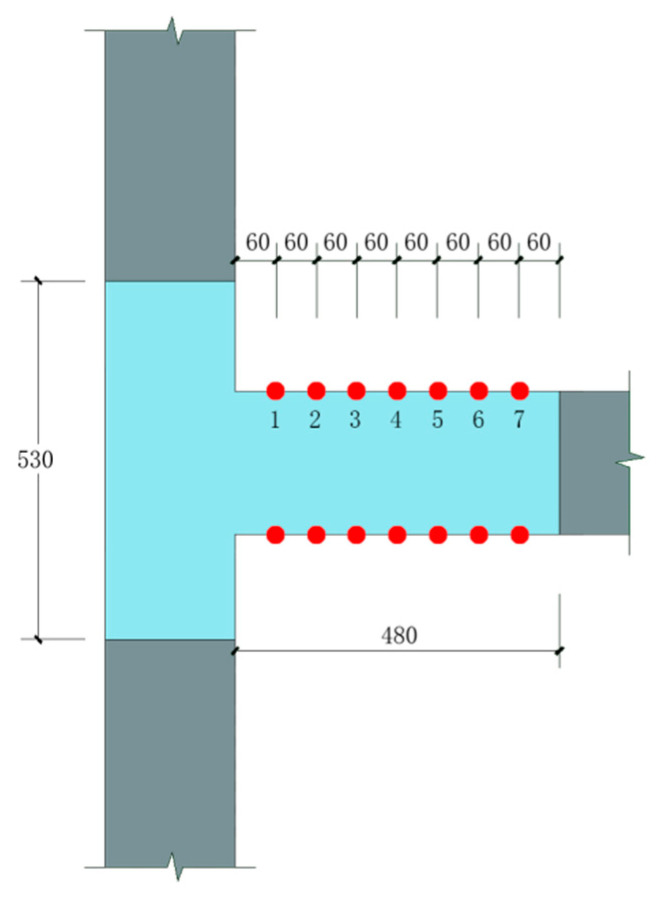
VIC-3D measuring area and the measuring point arrangement.

**Figure 9 materials-15-01704-f009:**
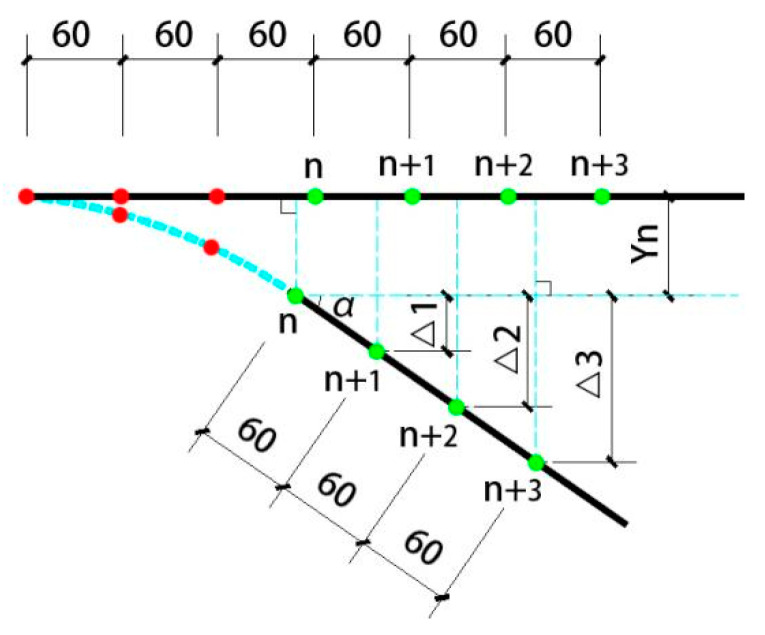
Schematic diagram of the plastic hinge length judgment principle (unit: mm).

**Figure 10 materials-15-01704-f010:**
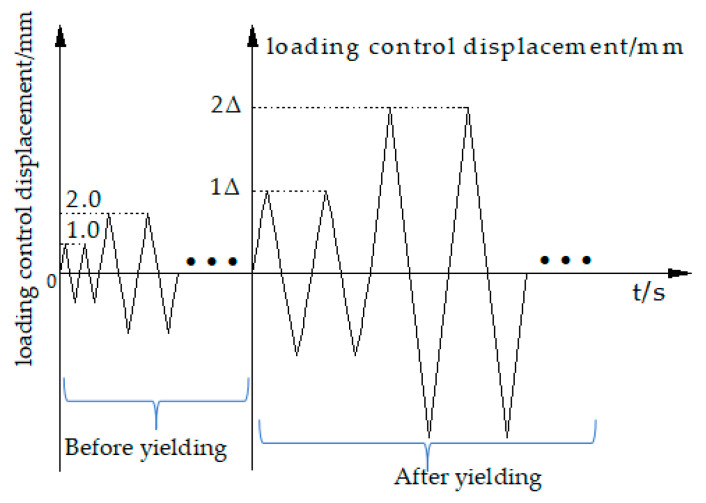
Loading protocol.

**Figure 11 materials-15-01704-f011:**
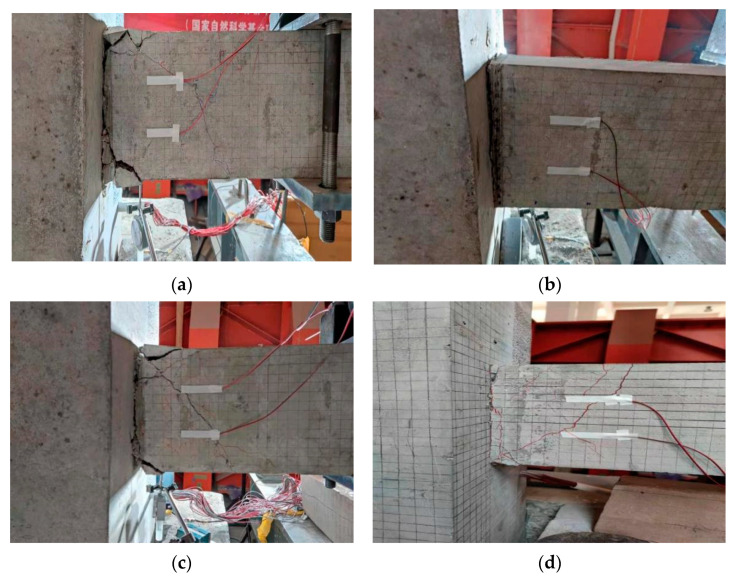
Failure modes of the specimens: (**a**) PJD-1; (**b**) SJD-1; (**c**) PSJD-1; (**d**) PSJD-2.

**Figure 12 materials-15-01704-f012:**
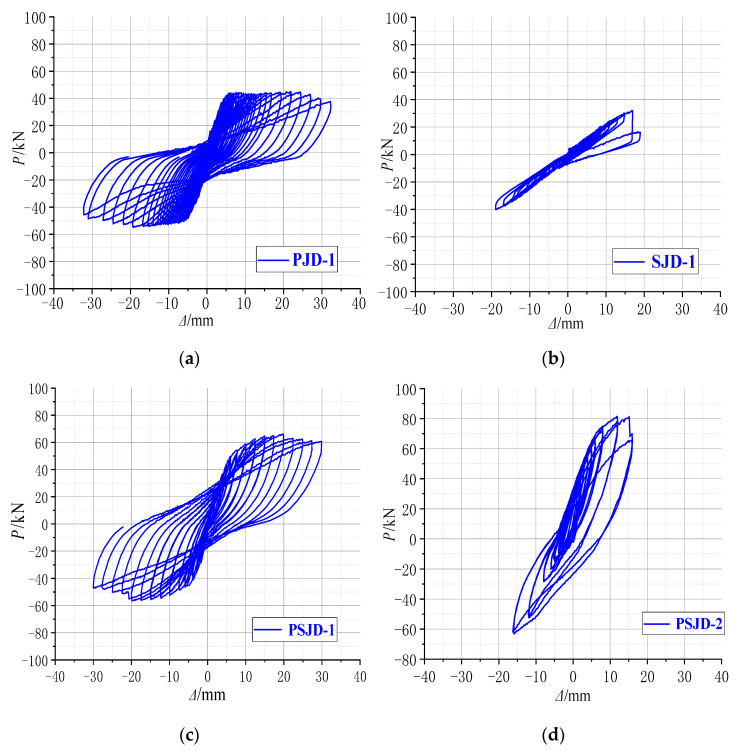
Hysteresis loops of the joint specimens: (**a**) PJD-1; (**b**) SJD-1; (**c**) PSJD-1; (**d**) PSJD-2.

**Figure 13 materials-15-01704-f013:**
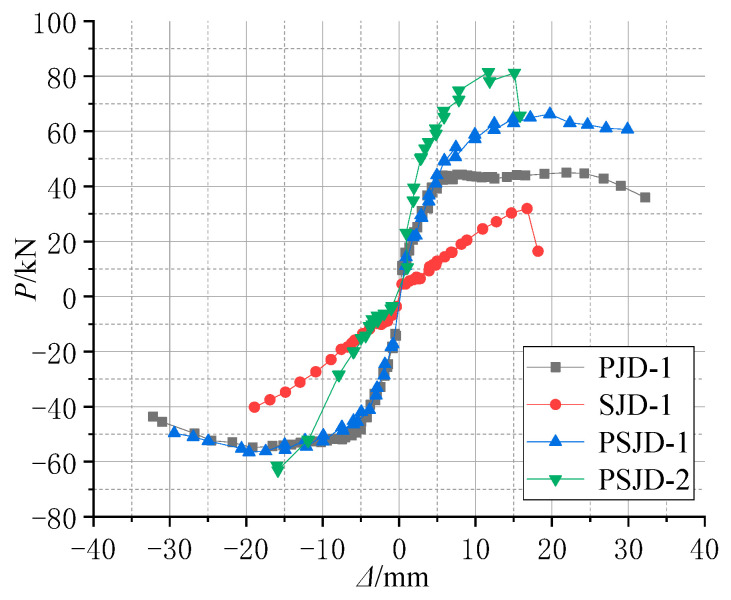
Envelope curves of the specimens.

**Figure 14 materials-15-01704-f014:**
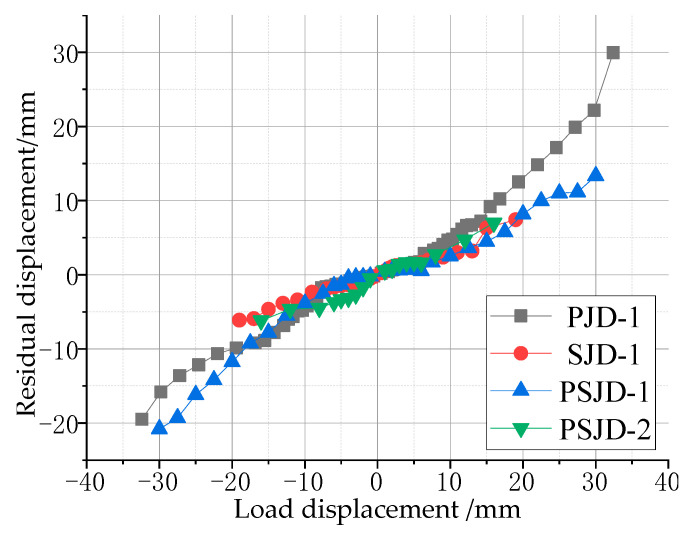
Residual displacement of the beam end after unloading.

**Figure 15 materials-15-01704-f015:**
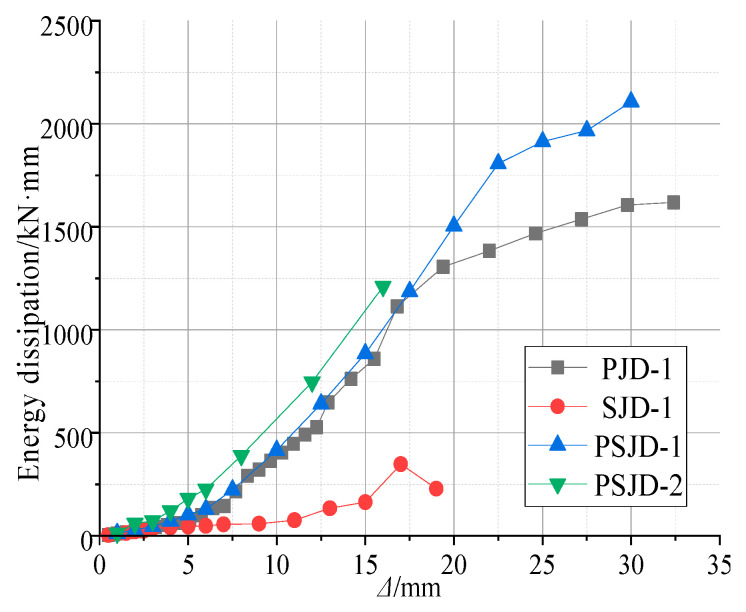
Energy dissipation capacity.

**Figure 16 materials-15-01704-f016:**
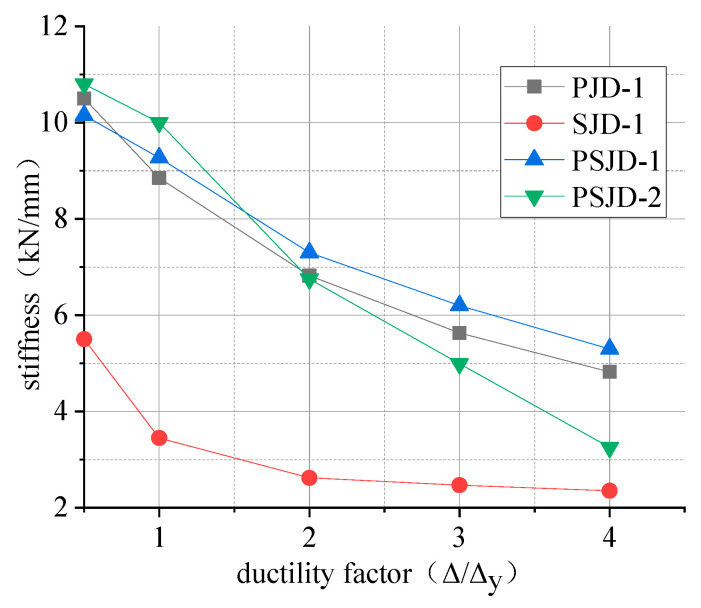
Curves of stiffness degradation.

**Figure 17 materials-15-01704-f017:**
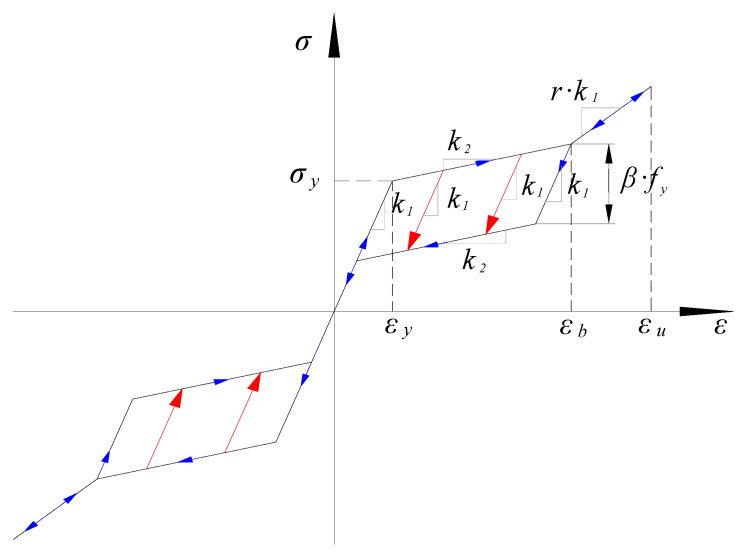
Self-centering constitutive model.

**Figure 18 materials-15-01704-f018:**
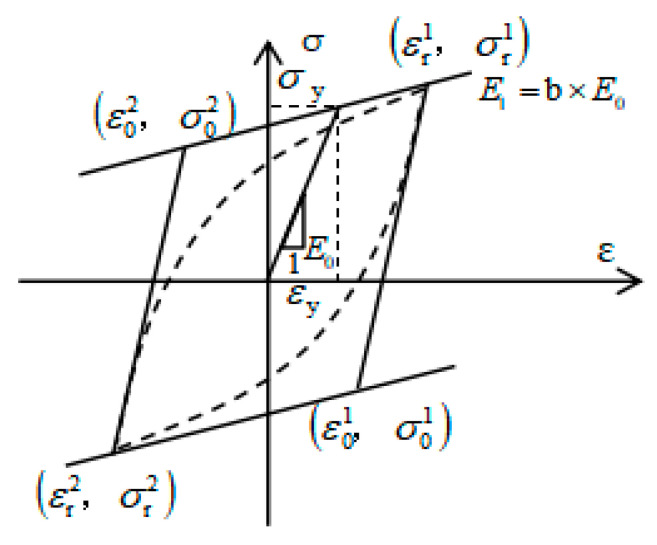
Steel 02 constitutive model.

**Figure 19 materials-15-01704-f019:**
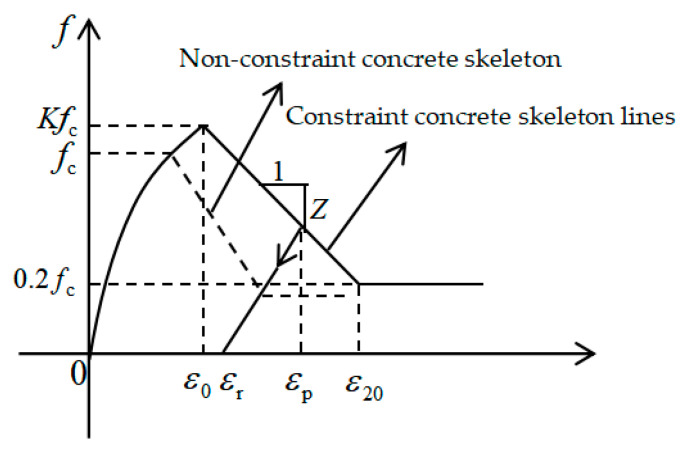
Concrete 01 constitutive model.

**Figure 20 materials-15-01704-f020:**
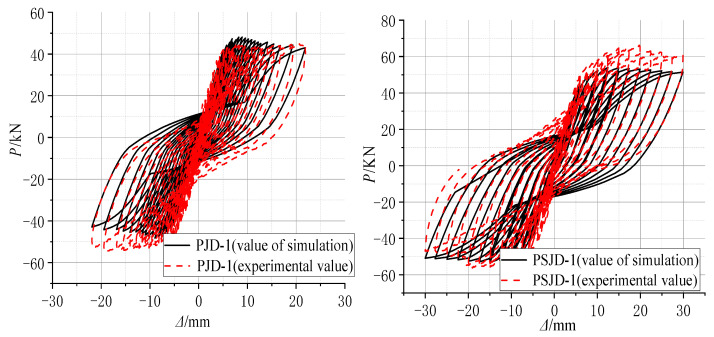
Comparison diagram of numerical simulation and experimental hysteresis curves.

**Figure 21 materials-15-01704-f021:**
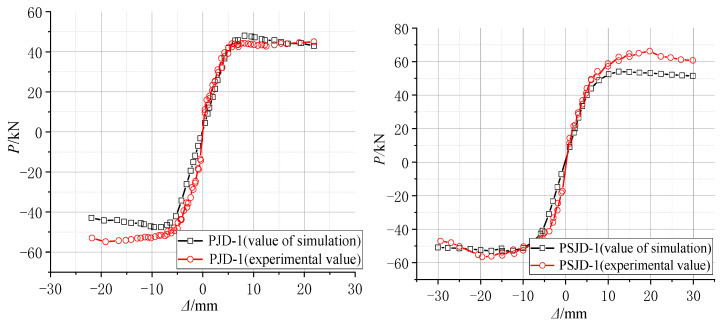
Comparison diagram of numerical simulation and experimental skeleton curves.

**Figure 22 materials-15-01704-f022:**
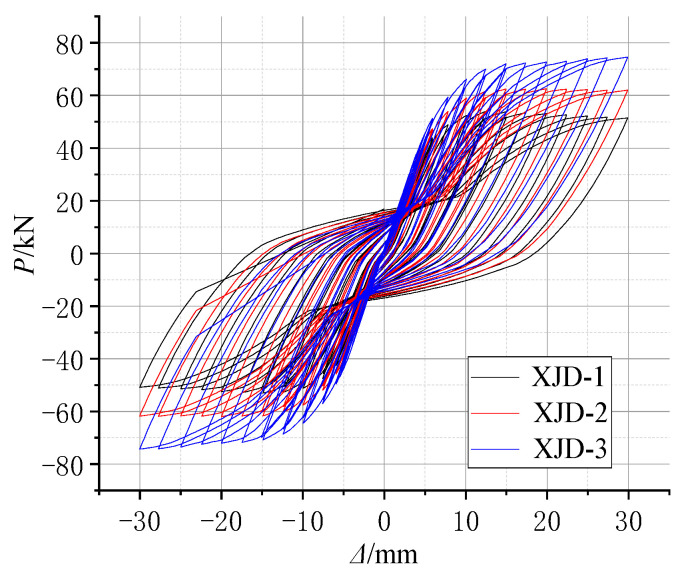
Hysteretic force–displacement curve of different diameters.

**Figure 23 materials-15-01704-f023:**
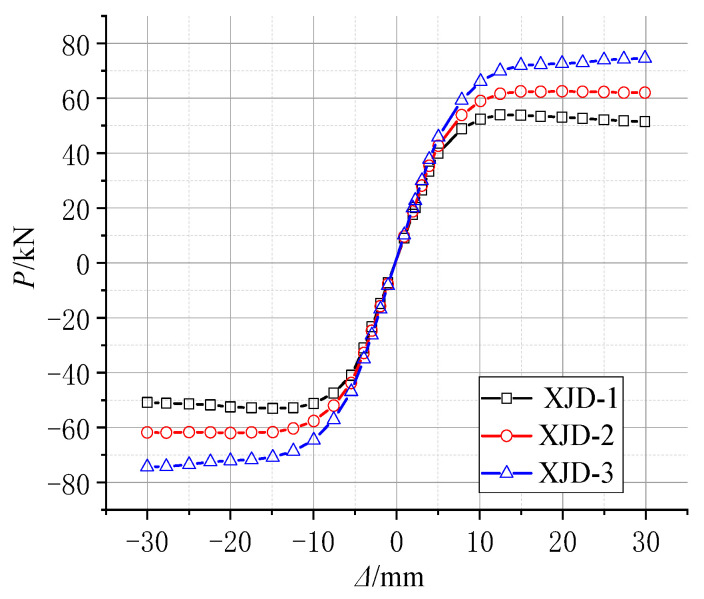
The skeleton curve of different diameters.

**Figure 24 materials-15-01704-f024:**
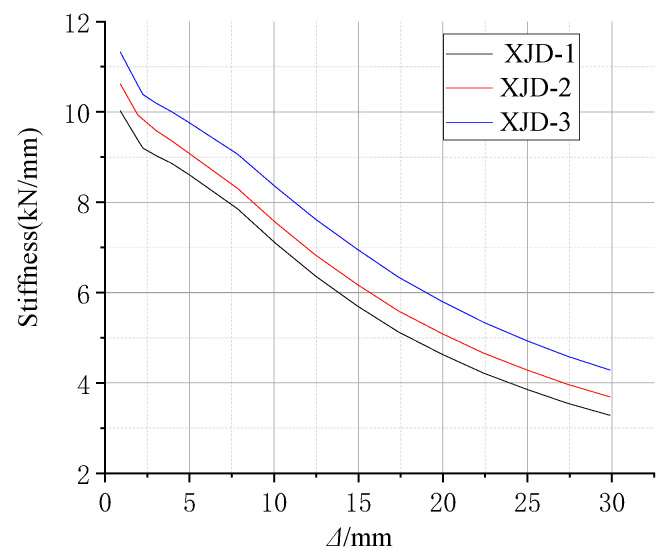
Stiffness degradation curve of different diameters.

**Figure 25 materials-15-01704-f025:**
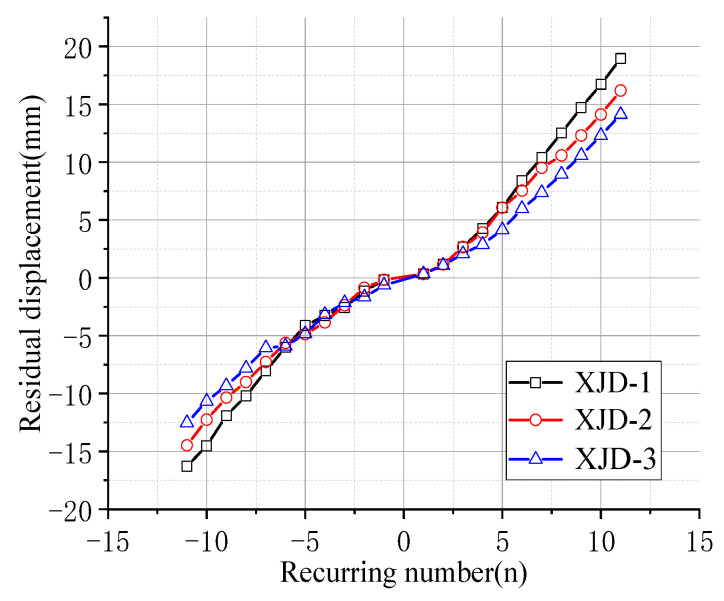
Residual deformation curves with different diameters.

**Figure 26 materials-15-01704-f026:**
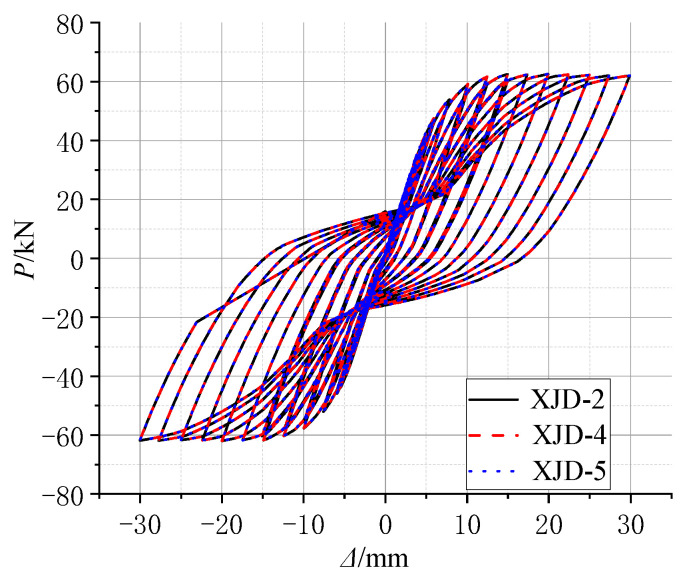
Comparison of hysteresis curves with different axial compression ratios.

**Table 1 materials-15-01704-t001:** Basic parameters of specimens.

Specimen Number	Column Reinforcement		Beam Reinforcement		Axial Compression Ratio	Test Purpose
	Longitudinal Reinforcement	Hooped Reinforcement	Longitudinal Reinforcement	SMA Reinforcement Ratio (%)		
PJD-1	4 × C22	A8@65/100	4 × C10	0	0.25	contrast test
SJD-1	4 × C22	A8@65/100	4 × SMA bars (10 mm)	0.800	0.25	contrast test
PSJD-1	4 × C22	A8@65/100	4 × C10 + 4 × SMA bars (8 mm)	0.513	0.25	model test
PSJD-2	4 × C22	A8@65/100	4 × C10 + 4 × SMA bars (10 mm)	0.800	0.25	model test

**Table 2 materials-15-01704-t002:** Material properties of the Ni–Ti alloy.

Material Name	Density ( $\mathrm{kg}\cdot m^{-3}$)	Young’s Modulus (GPa)	Tensile Strength (MPa)	Yield Strength (MPa)	Restoration Strain (%)
Ni–Ti	7800	65.4	600	390	4.5

**Table 3 materials-15-01704-t003:** The performance of the reinforcement.

Type of Steel Bar	Steel Bar Diameter d (mm)	Yield Strength $f_{y}$ (MPa)	Ultimate Strength $f_{u}$ (MPa)	Young’s Modulus (MPa)	Elongation Rate δ (%)
HPB300	6	310.67	460.56	1.90 × 10^5^	19.12
	8	323.98	462.73	1.92 × 10^5^	19.38
HRB400	10	448.56	601.09	2.03 × 10^5^	19.96
	22	438.64	573.86	1.98 × 10^5^	20.60

**Table 4 materials-15-01704-t004:** Concrete test block measurement results.

Measurement Items of Concrete Test Block	First Group			Second Heat			End Value
	1	2	3	4	5	6	
Failing load (kN)	543.13	436.56	455.34	485.01	411.73	470.97	
Average value (kN)	445.95			455.90			450.93
Compression strength (MPa)	42.37			43.31			42.84

**Table 5 materials-15-01704-t005:** Measured and calculated value of the plastic hinge length of each node.

Node Number	PJD-1	SJD-1	PSJD-1	PSJD-2
Y1 (mm)	8.13	2.11	9.20	0.71
Y2 (mm)	9.59	3.82	11.37	2.19
Y3 (mm)	11.36	5.52	13.73	3.72
Y4 (mm)	13.15	7.22	15.63	5.26
Y5 (mm)	14.93	8.92	17.74	6.81
Y6 (mm)	16.66	10.63	19.89	8.35
Y7 (mm)	18.39	—	22.05	—
sin*α* (×10^−^^2^)	2.90	2.80	3.60	2.50
Plastic hinge end number	5	1	5	4
Plasticity hinge length d (mm)	300	60	300	240

**Table 6 materials-15-01704-t006:** SMA simplified constitutive model parameters and values.

Parameter	Physical Significance	Value
$k_{1}$	First stiffness (N/mm)	75,000
$k_{2}$	Second stiffness (N/mm)	1827
$\sigma_{y}$	Positive phase transition stress (MPa)	400
β	Inverse phase transition stress coefficient	0.80
$\varepsilon_{s}$	Sliding strain	0.06
$\varepsilon_{b}$	Hardening strain	0.06
r	Hardening stiffness coefficient	0.39

**Table 7 materials-15-01704-t007:** Model parameter analysis scheme.

Node Trial Number	SMA Bar Diameter (mm)	Axial Compression Ratio
XJD-1	8	0.25
XJD-2	10	0.25
XJD-3	12	0.25
XJD-4	10	0.3
XJD-5	10	0.35

## Data Availability

The data used to support the finding of this study are available from the corresponding author upon request.

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
