# Peer review of "The Seismic Performance of New Self-Centering Beam-Column Joints of Conventional Island Main Buildings in Nuclear Power Plants"

_materials, 2022, doi:10.3390/ma15051704_

Round 1

Reviewer 1 Report

The paper has a great interest to readers due to the few experimental programs of BCj equipped with SMA materials. However, the manuscript must be revised to improve the state-of-the-art review which is weak, the presentation of figures, explanation of experimental program set up and results. Moreover, authors must check against all the occurrences of spaces not required, capital letters that are present when not needed and the contrary, ad so on. A comprehensive spelling revision is necessary. About grammar and phrasing I do not feel qualified to make suggestions but it could need some revision. Many figures must be improved to remove text overlapping of other issues. See also the attached pdf file

  1. the abstract is difficult to read. Please revise to make it clearer and more straightforward
  2. start of page 2: a brief reference is necessary to the fact that it is not economically convenient designing structural elements beyond seismic events that cause higher order problems like tsunami floods
  3. “and dissipate a lot of energy in this process” a lot of energy is an arbitrary judgment, sounding not scientific
  4. Literature review: while some reference to shape memory alloy studies is made, the author state that they want to apply SMA to beam-column joint without any reference to studies on the reinforced concrete beam-column joints seismic performances. In this direction it is necessary to add some literature references to empower the introduction making it sufficiently updated with respect to the state of the art, with reference to: general seismic behavior, influence of other structural components that are not present in experiments and other strengthening techniques.
    -A. Masi, G. Santarsiero (2013). Seismic tests on RC building exterior joints with wide beams. Proc. of the 2nd International Symposium on Materials Science and Engineering Technology (ISMSET 2013, June 27-28, 2013, Guangzhou, China). Advanced Material Reasearch. Vol 787: 771-777.
    -Santarsiero, G., Masi, A. (2020) Analysis of slab action on the seismic behavior of external RC beam-column joints. Journal of Building Engineering, Volume 32, November 2020, 101608
    https://doi.org/10.1016/j.jobe.2020.101608
    -Realfonzo R, Napoli A, Ruiz Pinilla JG (2014) Cyclic behavior of RC beam–column joints strengthened with FRP Systems. Constr Build Mater 54:282–297. https://doi.org/10.1016/j.conbuildmat.2013.12.043

  5. Section 1: a better name would be “Experimental program”, section 1.1 better “Test specimens”
  6. Section 1.1.: authors did not state that they are dealing with exterior joints. This must be specified due to the huge behavioural difference with internal joints.
  7. Scale ratio 1:5 is very small, maybe too small: please explain if you did something to make the materials suitable to such a small scale
  8. Improve fig. 2: labels seem to be of different font or height
  9. 2 it is not a “Structural photo” but a 3D sketch
  10. Fig 2: it seems that the steel bars are not bent into the joint core of the columns. Take into account that this configuration is very important to the BC joint. Please better scecify the shape of steel bars in the joint core.
  11. Title of Section 1.2 “Materials” is better
  12. Steel Q235: please specify to which code this name refers
  13. Fig 5 could be improved: 3D shape of holes does not look good. Dimensions are different each other
  14. Concrete testing: the test cubes dimensions are not reported
  15. Steel and concrete testing: testing protocols are not stated
  16. Fig 7 does not show the lateral bracing of fig. 6. Have you a better photo of the test set-up?
  17. Title of Section 1.5 “Loading history” would be better. Moreover, authors do not report the loading speed (e.g., mm/s) or do they adopt a different strategy for example cycles per second?
  18. Section 1.5. numbered list of paragraph seems to be not needed. Explain better the meaning of “before” and “after submission”, improve the figure since the text overlaps.
  19. Figure 13. Graphs need to be made with same force (100 kN) and displ.(40mm) Scale. Set on the maximum to make appreciate the difference between specimens

Author Response

Please see the attachment, thank you!

Reviewer 2 Report

The paper in general presents interesting and practical subject and contains both experimental and analytical parts and discussions. It has been well written and organized and the reviewer believes that it can be published. However, some comments needs to be answered in the revised version. - Some abbreviations such as “RC” and “SMA” in the abstract should be defined. - Please add a list of nomenclature - In the introduction part more descriptions is needed for explaining the benefits and advantages of the new type joint compared to the other conventional and available methods - The list of References at the end of manuscript shows that almost all of the cited works are chosen from a limited group of scholars and a specific country. It is suggested citing other papers from other research communities as well. some related works are: “Engineering Solid Mechanics 6(4), 341-352; Engineering Solid Mechanics 7(3), 247-262; Engineering Solid Mechanics 6(4), 331-340;”. - It seems that the abbreviation “SMA” has been used for defining both “Shape Memory Alloy” and “Seismic Margin Assessment”. This issue must be edited in the revised version. - More details about the experimental procedure and gages shown in Fig. 12 should be explained. - Please explain What was the criterion for the onset of failure in the experiments - In Page 16 after Eq. (4), please revise a written “Chinese word”

Author Response

Please see the attachment, thank you!

Reviewer 3 Report

The paper has been written well and can be considered for publication. Some minor comments to improve the quality of the paper.

  1. The numbering of the Introduction was ‘0’. The numbering of a paragraph should start from ‘1’. Change the numbering of the Introduction to ‘1’ and update the numbering for other paragraphs accordingly.
  2. Change the title of the paragraph ‘Test Introduction’ to something ‘Test Program’ or ‘Experimental Program’ for better readability.
  3. In Section ‘Experimental Model’ some sentences are written in the present tense where others are in the past tense. Be consistent. For example, the sentence ‘The geometric dimensions of each specimen are the same and are made according to the current concrete design specifications in China.’ Throughout the paper, use past tense for describing the test program.
  4. Section 1.2, starts the word with the capital alphabet. Same comment for section 2.3
  5. The following papers should cite in support of the computational advancement of the fiber model: (a) Engineering Structures 2018;175:13-26. (b) Thin-Walled Structures 2019;138:105-116. (c) Engineering Structures 2017;131:639-650.

Author Response

Please see the attachment, thank you!

Round 2

Reviewer 2 Report

I recommend publishing this revised version